# Effects of low-dose pirfenidone on survival and lung function decline in patients with idiopathic pulmonary fibrosis (IPF): Results from a real-world study

**Eung Gu Lee**[1]**, Tae-Hee Lee**[2]**, Yujin Hong**[1]**, Jiwon Ryoo**[1]**, Jung Won Heo**[3]**, Bo Mi Gil**[4]**, Hye Seon Kang**[1]**, Soon Seog Kwon**[1]**, Yong Hyun Kim** [1]*

**1** Division of Pulmonary, Allergy and Critical Care Medicine, Department of Internal Medicine, Bucheon St. Mary's Hospital, College of Medicine, The Catholic University of Korea, Seoul, Republic of Korea, **2** Department of Statistics and Data Science, Yonsei University, Seoul, South Korea, **3** Division of Pulmonary, Critical Care and Sleep Medicine, Department of Internal Medicine, Eunpyeong St. Mary's Hospital, College of Medicine, The Catholic University of Korea, Seoul, Republic of Korea, **4** Department of Radiology, Bucheon St. Mary's Hospital, College of Medicine, The Catholic university of Korea, Seoul, Republic of Korea

* kyh30med@catholic.ac.kr

## Abstract

### Background

Idiopathic pulmonary fibrosis (IPF) is a chronic, progressive fibrosing interstitial pneumonia of unknown etiology. In several randomized clinical trials, and in the clinical practice, pirfenidone is used to effectively and safely treat IPF. However, sometimes it is difficult to use the dose of pirfenidone used in clinical trials. This study evaluated the effects of low-dose pirfenidone on IPF disease progression and patient survival in the real-world.

### Methods

This retrospective, observational study enrolled IPF patients seen at the time of diagnosis at a single center from 2008 to 2018. Longitudinal clinical and laboratory data were prospectively collected. We compared the clinical characteristics, survival, and pulmonary function decline between patients treated and untreated with various dose of pirfenidone.

### Results

Of 295 IPF patients, 100 (33.9%) received pirfenidone and 195 (66.1%) received no antifibrotic agent. Of the 100 patients who received pirfenidone, 24 (24%), 50 (50%), and 26 (26%), respectively, were given 600, 1200, and 1800 mg pirfenidone daily. The mean survival time was 57.03 ± 3.90 months in the no-antifibrotic drug group and 73.26 ± 7.87 months in the pirfenidone-treated group (p = 0.027). In the unadjusted analysis, the survival of the patients given pirfenidone was significantly better (hazard ratio [HR] = 0.69, 95% confidence interval [CI]: 0.48–0.99, p = 0.04). After adjusting for age, gender, body mass index, and the GAP score [based on gender (G), age (A), and two physiological lung parameters (P)], survival remained

**Funding:** The author(s) received no specific funding for this work.

**Competing interests:** The authors have declared that no competing interests exist.

**Abbreviations:** IPF, idiopathic pulmonary fibrosis; HR, hazard ratio; CI, confidence interval; BMI, body mass index; FVC, forced vital capacity; FEV1, forced expiratory volume in 1 second; DLCO, diffusing capacity of lung for carbon monoxide; TGF-β, tumor growth factor-β; TNF-α, tumor necrosis factor-α; AEs, adverse events; GI, gastrointestinal; OS, overall survival; ATS, American Thoracic Society; ERS, European Respiratory Society; JRS, Japanese Respiratory Society; ALAT, Latin American Thoracic Association; GAP, gender (G), age (A), and two physiological lung parameters (P), i.e., the FVC and DLCO; SD, standard deviation; SE, standard error; 6MWT, 6-minute walk test; SpO2, saturation of percutaneous oxygen; BAL, bronchoalveolar lavage; BSA, Body surface area.

better in the patients given pirfenidone (HR = 0.56, 95% CI: 0.37–0.85, p = 0.006). In terms of pulmonary function, the decreases in forced vital capacity (%), forced expiratory volume in 1 s (%) and the diffusing capacity of lung for carbon monoxide (%) were significantly smaller (p = 0.000, p = 0.001, and p = 0.007, respectively) in patients given pirfenidone.

## Conclusions

Low-dose pirfenidone provided beneficial effects on survival and pulmonary function decline in the real-world practice.

## Introduction

Idiopathic pulmonary fibrosis (IPF) is a chronic, progressive fibrosing interstitial pneumonia of unknown etiology [1]. IPF is fatal, characterized by progressive dyspnea and irreversible loss of lung function. The disease course is variable and unpredictable, and the median survival time from diagnosis is 2–4 years [2]. The pathophysiology of IPF is characterized by recurrent epithelial cell injury, senescence of alveolar epithelial cells, the expression of profibrotic mediators that stimulate matrix deposition by myofibroblasts, microbiome changes, and abnormalities in host defense [3]. Two anti-fibrotic drugs, pirfenidone and nintedanib, are used to effectively and safely treat IPF [4]. Pirfenidone is an orally administered pyridine with anti-inflammatory, anti-oxidant, and anti-fibrotic actions. The drug inhibits collagen synthesis, downregulates of expression of tumor growth factor (TGF)-β and tumor necrosis factor (TNF)-α, and reduces fibroblast proliferation [5]. In the CAPACITY and ASCEND trials, pirfenidone at 2403 mg/day (801 mg three times daily) reduced disease progression reflected in lung function, exercise tolerance, and progression-free survival, in IPF patients compared to a placebo [6, 7]. Pirfenidone was approved in Europe in 2011, and in the United States in 2014. The most recent clinical practice guideline for IPF includes conditional recommendations for its use in most patients [4]. The most common adverse events (AEs) observed in clinical trials and the real-world experience are gastrointestinal (GI) and skin-related; they are generally mild-to-moderate and rarely lead to treatment discontinuation [6–10]. Although long-term pirfenidone is generally well-tolerated, dose modification can reduce both the incidence and severity of AEs, and promote patient compliance [11, 12].

As a complement to clinical trials, real-world studies have less strict inclusion criteria [13]. Several studies have evaluated the real-world efficacy and safety of pirfenidone, which was found to delay disease progression [8, 13, 14]. In a phase III clinical trial, Japanese patients given low-dose pirfenidone (1200 mg/day) showed a significantly slower decline in forced vital capacity (FVC) compared to the placebo group [9]. Although real-world data are gradually accumulating, more longitudinal data on IPF disease progression and survival in patients on low-dose pirfenidone (≤1200 mg/day) are required [8]. This study compared the clinical characteristics, overall survival (OS), and pulmonary function decline between patients treated and untreated with low-dose pirfenidone at a single institution.

## Methods

### Study design and population

This single-center, retrospective observational study reviewed the medical records and laboratory test data of patients diagnosed with IPF at Bucheon St. Mary's hospital, South Korea. IPF

patients consecutively enrolled at diagnosis from 2008 to 2018 were evaluated. Patients' characteristics (age, gender, smoking status, BMI) and clinical characteristics (medical history, diagnosis, pulmonary function, radiologic patterns, biomarkers) were collected. Clinical and laboratory data, including pulmonary function test and image studies were collected regularly and in real-time at the time of workup and follow up by the pre-set protocols specified for ILD. All the data collected were again retrospectively reviewed. Data on medications were collected throughout the study, including immunosuppressive agents.

The radiological images and pathological findings were evaluated by pulmonologists, radiologists, and pathologists. All IPF diagnoses were reconfirmed according to the Official American Thoracic Society (ATS)/European Respiratory Society (ERS)/Japanese Respiratory Society (JRS)/Latin American Thoracic Association (ALAT) Clinical practice guideline, 2018 [1]. The GAP index and a staging system are used to predict the clinical prognosis of IPF patients. The GAP index evaluates gender (G), age (A), and two physiological lung parameters (P), i.e., the FVC and diffusing capacity of lung for carbon monoxide (DLCO). Patients were staged as I–III, for which the estimated 1-year mortality rates are 5.6%, 16.2% and 39.2%, respectively [15].

There is a compulsory and universal health insurance system in South Korea. Pirfenidone is an expensive drug and was approved by the health insurance system in October 2015. The reimbursement criteria for pirfenidone are strict and are limited to patients with a definite IPF based on high resolution CT and/or surgical lung biopsy with FVC $\leq$ 90% or DLCO $\leq$ 80%. Therefore, since then, pirfenidone has been established as the standard-of-care for patients who satisfy the above criteria, and is also used in our hospital. However, nintedanib is not approved to reimburse and is rarely prescribed due to its high cost in South Korea.

Pirfenidone was initially administered with food as three daily 200 mg doses, and then gradually increased to the full dose of 1800 mg/day over every 2–4 weeks in daily increments of 200–600 mg. The patient's condition was carefully monitored during this process. Patients who could not tolerate the full recommended dose (1800 mg/day) due to AEs were assigned to the low-dose (600 or 1200 mg/day) pirfenidone group. Dose escalation, dose reduction, or discontinuation of pirfenidone was made at the physician's discretion, considering the patients' condition and not for research purposes.

## Statistical analysis

Data were expressed as the mean ± standard deviation (SD) or the mean ± standard error (SE), or as numbers with percentages, as appropriate. A student's *t*-test was used to compare continuous variables between the groups, and the Pearson chi-squared test or ANOVA was used to compare of categorical variables. Survival probabilities were estimated using the Kaplan-Meier method. Adjusted hazard ratios (aHRs) and 95% confidence intervals (CIs) were calculated using a Cox proportional hazards model adjusted for age, gender, body mass index (BMI), and the GAP score. The statistical analyses were performed using SAS software (ver. 9.4; SAS Institute, Cary, NC, USA), except for the survival analysis, which was done using SPSS Statistics for Windows software (ver. 24.0; IBM Corp., Armonk, NY, USA). A two-sided p-value $\leq$ 0.05 was taken to indicate a significant difference.

## Ethics statement

The study was approved by the Institutional Review Board (IRB) and Ethics Committee of Bucheon St. Mary's Hospital (IRB No.: 2021-3027-0001). The need for written informed consent was waived because of the retrospective design.

## Results

### Demographics

Baseline subject characteristics are presented in Table 1. Of the 295 patients diagnosed with IPF, 195 did not use any antifibrotic drug and 100 were taking pirfenidone. The mean age was 70.81 ± 10.68 years in the no-antifibrotic drug group and 68.87 ± 8.48 years in the pirfenidone-treated group (p = 0.099). The proportions of males (81.0% vs. 61.5%, p = 0.0007) and current or former smokers (75.0% vs. 57.4%, p = 0.003) were higher in the pirfenidone-treated group than in the no-antifibrotic drug group; there was no significant group difference in BMI (23.53 ± 3.35 vs. 23.71 ± 3.49 kg/m$^2$, p = 0.675).

The forced expiratory volume in 1 second (FEV1) was significantly higher in the pirfenidone-treated group than the no-antifibrotic drug group (2.21 ± 0.60 vs. 2.04 ± 0.63 L, p = 0.039). Other pulmonary function parameters, including the FVC, FEV1/FVC ratio, and DLCO did not differ between the two groups.

The distance covered in the 6-minute walk test (6MWT) did not differ between the no-antifibrotic drug and pirfenidone-treated groups (368.0 ± 179.27 vs. 391.72 ± 157.47 m, p = 0.411),

**Table 1. Baseline epidemiological and clinical characteristics of the enrolled patients.**

| Characteristics | No-antifibrotic drug (n = 195) | Pirfenidone (n = 100) | p-value |
|---|---|---|---|
| Age, year | 70.81 ± 10.68 | 68.87 ± 8.48 | 0.099 |
| Male, n (%) | 120 (61.5%) | 81 (81.0%) | 0.0007 |
| BMI, kg/m$^2$ | 23.71 ± 3.49 | 23.53 ± 3.35 | 0.675 |
| Current or former smokers, n (%) | 112 (57.4%) | 75 (75.0%) | 0.003 |
| Pack years | 21.44 ± 24.9 | 25.65 ± 21.01 | 0.151 |
| Bronchoalveolar lavage (BAL), n (%) | 94 (48.2%) | 71 (71.0%) | 0.001 |
| Surgical lung biopsy, n (%) | 44 (22.6%) | 12 (12.0%) | 0.029 |
| 6 minute walk test (6MWT), n (%) | 368 ± 179.27 | 391.72 ± 157.47 | 0.411 |
| SpO2 ≥90% after 6MWT, n (%) | 153 (78.5%) | 42 (42.0%) | 0.0001 |
| GAP score | 2.91 ± 1.25 | 3.27 ± 1.35 | 0.0251 |
| Stage, n (%) | | | 0.242 |
| I | 144 (73.9%) | 65 (65.0%) | |
| II | 43 (22.0%) | 31 (31.0%) | |
| III | 8 (4.1%) | 4 (4.0%) | |
| Chest CT pattern, n (%) | | | |
| UIP | 140 (71.8%) | 70 (70.0%) | |
| Probable UIP | 31 (15.9%) | 23 (23.0%) | |
| Indeterminate UIP | 24 (12.3%) | 7 (7.0%) | |
| Pulmonary function test (PFT) | | | |
| FVC (L) | 2.54 ± 0.83 | 2.70 ± 0.76 | 0.122 |
| FVC (% predicted) | 81.73 ± 18.97 | 79.84 ± 18.99 | 0.435 |
| FEV1 (L) | 2.04 ± 0.63 | 2.21 ± 0.60 | 0.039 |
| FEV1 (% predicted) | 96.2 ± 24.74 | 95.35 ± 25.03 | 0.789 |
| FEV1/FVC | 81.49 ± 8.79 | 82.35 ± 7.02 | 0.377 |
| DLCO (mL/mmHg/min) | 11.13 ± 5.47 | 10.74 ± 4.46 | 0.536 |
| DLCO (% predicted) | 67.23 ± 24.95 | 64.01 ± 24.75 | 0.314 |

Values are expressed as mean ± standard deviation (SD).

Abbreviations: Standard deviation, SD; Body mass index, BMI; Bronchoalveolar lavage, BAL; 6-minute walk test (6MWT), Usual interstitial pneumonia, UIP; Pulmonary function test, PFT; Forced vital capacity, FVC; Forced expiratory volume in one second, FEV1; Diffusing capacity of lung for carbon monoxide, DLCO.

but the proportion of patients with a saturation of percutaneous oxygen (SpO2) percentage over 90% after the 6MWT was significantly higher in the former group (78.5% vs. 42.0%, p = 0.025).

The severity of IPF, as evaluated by the GAP index, was greater in the pirfenidone-treated than the no-antifibrotic drug group (3.27 ± 1.35 vs. 2.91 ± 1.25, p = 0.025). However, there was no significant difference between the two groups in the proportions of GAP stage I–III patients. The proportion of patients who underwent bronchoalveolar lavage (BAL) at the time of diagnosis was significantly higher in the pirfenidone-treated than no-antifibrotic drug group (71.0% vs. 48.2%, p = 0.001), while the proportion diagnosed with IPF via surgical lung biopsy was significantly higher in the latter group (12.0% vs. 22.6%, p = 0.029).

Of the 100 patients treated with pirfenidone, 24 received 600 mg/day (24.0%), 50 received 1200 mg/day (50.0%), and 26 received 1800 mg/day (26.0%) (Table 2). The mean age (70.12 ± 8.90, 70.52 ± 8.10 and 65.00 ± 7.91 years, respectively, p = 0.021) was significantly lower, and the BMI was significantly higher (22.04 ± 3.04, 23.49 ± 2.95 and 24.99 ± 3.81 kg/m$^2$, respectively, p = 0.007), in the group receiving 1800 mg/day of pirfenidone.

## Survival analysis

The mean OS was 57.03 ± 3.90 months in the no-antifibrotic drug group and 73.26 ± 7.87 months in the pirfenidone-treated group (p = 0.027; Fig 1). There was no significant difference in OS between patients given the full dose of pirfenidone recommended in South Korea (1800 mg/day) and those treated with lower doses (600 or 1200 mg/day) (Fig 2). The mean survival time was 73.26 ± 10.12 and 72.96 ± 9.75 months in patients treated with the full and lower doses, respectively (p = 0.603).

**Table 2. Baseline epidemiological and clinical characteristics according to dose in patients treated with pirfenidone (n = 100).**

| Characteristics | Pirfenidone 600 mg/day (n = 24) | Pirfenidone 1200 mg/day (n = 50) | Pirfenidone 1800 mg/day (n = 26) | p-value |
|---|---|---|---|---|
| Age, year | 70.12 ± 8.90 | 70.52 ± 8.10 | 65.00 ± 7.91 | 0.021 |
| Male, n (%) | 18 (75.0%) | 40 (80.0%) | 23 (88.5%) | 0.925 |
| BMI, kg/m$^2$ | 22.04 ± 3.04 | 23.49 ± 2.95 | 24.99 ± 3.81 | 0.007 |
| BSA, m$^2$ | 1.60 ± 0.21 | 1.67 ± 0.25 | 1.74 ± 0.18 | 0.037 |
| Current or former smokers, n (%) | 18 (75.0%) | 35 (70.0%) | 22 (84.6%) | 0.377 |
| Pack years | 25.36 ± 20.85 | 24.88 ± 22.96 | 27.44 ± 17.59 | 0.884 |
| GAP score | 3.5 ± 1.47 | 3.36 ± 1.16 | 2.88 ± 1.56 | 0.223 |
| Stage, n (%) | | | | 0.386 |
| I | 13 (54.2%) | 33 (66.0%) | 19 (73.1%) | |
| II | 10 (41.7%) | 16 (32.0%) | 5 (19.2%) | |
| III | 1 (4.1%) | 1 (2.0%) | 2 (7.7%) | |
| Pulmonary function test (PFT) | | | | |
| FVC (L) | 2.66 ± 0.97 | 2.65 ± 0.74 | 2.83 ± 0.60 | 0.619 |
| FVC (% predicted) | 81.23 ± 22.26 | 79.88 ± 20.00 | 78.62 ± 14.10 | 0.895 |
| FEV1 (L) | 2.14 ± 0.78 | 2.19 ± 0.58 | 2.31 ± 0.49 | 0.593 |
| FEV1 (% predicted) | 96.82 ± 30.11 | 96.88 ± 26.74 | 91.31 ± 15.88 | 0.632 |
| FEV1/FVC | 80.82 ± 6.83 | 83.21 ± 6.91 | 82.08 ± 7.42 | 0.410 |
| DLCO (mL/mmHg/min) | 9.95 ± 5.16 | 10.40 ± 4.06 | 12.04 ± 4.42 | 0.205 |
| DLCO (% predicted) | 60.09 ± 23.76 | 64.23 ± 25.28 | 66.92 ± 25.11 | 0.637 |

Values are expressed as the mean ± standard deviation (SD).

Abbreviations: Standard deviation, SD; Body mass index, BMI; Body surface area, BSA; Pulmonary function test, PFT; Forced vital capacity, FVC; Forced expiratory volume in one second, FEV1; Diffusing capacity of lung for carbon monoxide, DLCO.

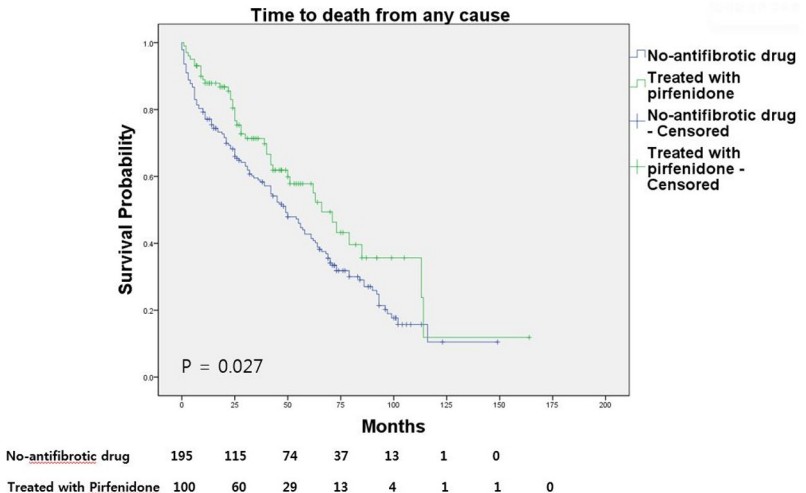

**Fig 1. Overall survival of IPF patients on pirfenidone and no-antifibrotic treatment.**

The association between pirfenidone treatment and mortality was analyzed (Table 3). Regardless of dose, the mortality rate was significantly lower in patients on pirfenidone (HR = 0.691, 95% CI: 0.484–0.986, p = 0.042). The analysis was repeated after adjusting for age, gender, BMI, and the GAP score; the mortality rate of patients treated with pirfenidone remained significantly lower (HR = 0.563, 95% CI: 0.374–0.845, p = 0.006). There was no significant association between the dose of pirfenidone (600 or 1200 vs. 1800 mg/day) and mortality (HR = 0.865, 95% CI: 0.421–1.779, p = 0.694), including adjusting for age, gender, BMI, and the GAP score (HR = 1.050, 95% CI: 0.472–2.338, p = 0.905).

## Pulmonary function

Pulmonary function was analyzed in IPF patients who had undergone at least two pulmonary function tests (87 patients who did not use antifibrotic drugs and 55 patients on pirfenidone).

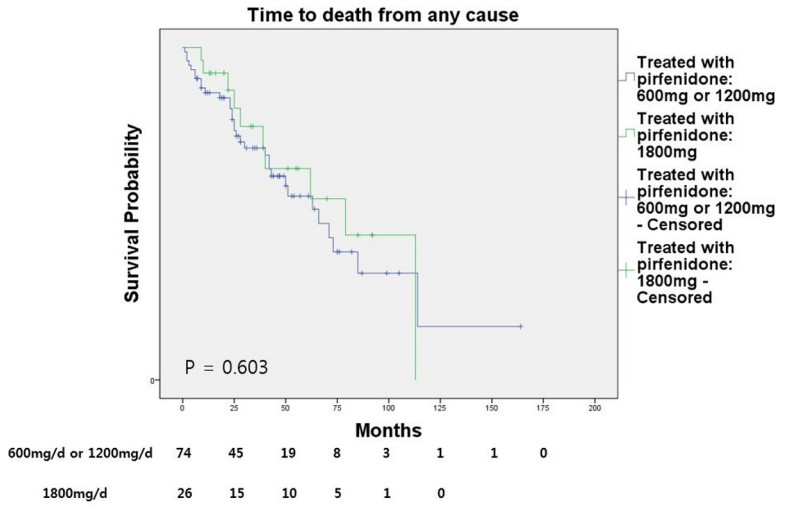

**Fig 2. Overall survival of IPF patients according to the pirfenidone dose–full recommended dose of 1800 mg/day vs. relatively low-dose of 1200 mg/day or less.**

**Table 3. Effects of pirfenidone treatment on mortality using Cox proportional hazard regression model.**

| | Unadjusted analysis | | | Adjusted analysis | | |
|---|---|---|---|---|---|---|
| | HR | 95% CI | P value | HR | 95% CI | P value |
| No-antifibrotic drug | 1 | | | 1 | | |
| Treated with pirfenidone | 0.691 | 0.484–0.986 | 0.042 | 0.563 | 0.374–0.845 | 0.006 |
| Relatively low dose pirfenidone (600mg/d and 1200mg/d) | 1 | | | 1 | | |
| Full recommended dose pirfenidone (1800mg/d) | 0.865 | 0.421–1.179 | 0.694 | 1.050 | 0.472–2.338 | 0.905 |

Abbreviations: Hazard ratio, HR; Confidence interval, CI.

Significantly smaller decreases in all pulmonary function indices, except the FEV1/FVC, were seen in patients treated with any dose (600, 1200, or 1800 mg/day) of pirfenidone than those not with the drug (Tables 4–6). There was no significant difference in all pulmonary function indices in patients treated with the full dose (1800 mg/day) of pirfenidone and lower dose (600 or 1200 mg/day) (Table 7).

### Adverse events

The AEs of patients on pirfenidone are summarized in Table 8. Most of the AEs affected the GI tract and skin. The proportion of patients who experienced at least one AE was significantly higher in the patient group treated with the full pirfenidone dose than the groups given the lower doses (44.6% vs. 92.3%, p = 0.000). However, the AE incidence, and the frequency and causes of pirfenidone discontinuation, did not differ between the two groups (Table 9). The proportion of patients with follow-up loss showed a tendency to be higher in the group of patients who were prescribed relatively low-dose pirfenidone. Still, the detailed reason could not be identified.

### Discussion

In South Korea, pirfenidone was approved by the Korean food and Drug Administration in 2012. However, due to the high price and the lack of clinical practice of pirfenidone, it was not included in the health insurance system until October 2015. Among enrolled patients, the majority of patients not treated with pirfenidone were diagnosed in the pre-antifibrotic era.

In addition, the reimbursement criteria for pirfenidone are strict and are limited to patients with a definite usual interstitial pneumonia pattern on high resolution CT or IPF diagnosed by

**Table 4. Comparison of pirfenidone treatment and annual decline in pulmonary function.**

| | Annual decline | | |
|---|---|---|---|
| | No-antifibrotic drug (n = 87) | Treated with pirfenidone (n = 55) | p-value |
| ΔFVC (L) | -0.328 ± 0.301 | -0.130 ± 0.367 | 0.001 |
| ΔFVC (% predicted) | -9.85 ± 11.432 | -1.548 ± 9.809 | 0.000 |
| ΔFEV1 (L) | -0.219 ± 0.244 | -0.032 ± 0.312 | 0.000 |
| ΔFEV1 (% predicted) | -9.786 ± 13.917 | 1.880 ± 25.670 | 0.001 |
| ΔFEV1/FVC | 1.343 ± 5.841 | 3.297 ± 16.983 | 0.326 |
| ΔDLCO (mL/mmHg/min) | -2.177 ± 2.975 | -0.572 ± 2.741 | 0.002 |
| ΔDLCO (% predicted) | -11.695 ± 16.819 | 0.765 ± 37.177 | 0.007 |

Values are expressed as the mean ± standard deviation (SD).

Abbreviations: Standard deviation, SD; Forced vital capacity, FVC; Forced expiratory volume in one second, FEV1; Diffusing capacity of lung for carbon monoxide, DLCO.

**Table 5. Comparison of annual decline in pulmonary function between patients who were not treated with antifibrotic drugs and patients who were treated with full recommended dose of pirfenidone (1800mg/d).**

| | Annual decline | | p-value |
|---|---|---|---|
| | No-antifibrotic drug (n = 87) | Treated with full recommended dose of pirfenidone (1800mg/d) (n = 19) | |
| ΔFVC (L) | -0.328 ± 0.301 | -0.081 ± 0.287 | 0.002 |
| ΔFVC (% predicted) | -9.85 ± 11.432 | -2.123 ± 7.683 | 0.001 |
| ΔFEV1 (L) | -0.219 ± 0.244 | -0.042 ± 0.268 | 0.014 |
| ΔFEV1 (% predicted) | -9.786 ± 13.917 | -0.963 ± 10.225 | 0.003 |
| ΔFEV1/FVC | 1.343 ± 5.841 | 1.066 ± 4.062 | 0.807 |
| ΔDLCO (mL/mmHg/min) | -2.177 ± 2.975 | -0.793 ± 2.533 | 0.045 |
| ΔDLCO (% predicted) | -11.695 ± 16.819 | -2.921 ± 13.123 | 0.018 |

Values are expressed as the mean ± standard deviation (SD).

Abbreviations: Standard deviation, SD; Forced vital capacity, FVC; Forced expiratory volume in one second, FEV1; Diffusing capacity of lung for carbon monoxide, DLCO.

surgical lung biopsy and with FVC ≤ 90% or DLCO ≤ 80%. Among patients diagnosed with IPF, patients diagnosed earlier than 2015 and did not meet the above pulmonary function criteria because their pulmonary function was preserved cannot be treated with pirfenidone.

The Official ATS/ERS/JRS/ALAT Clinical practice guideline in 2015 recommended not use combination therapy of N-acetylcysteine, azathioprine, and prednisone in patients with IPF. Previously, immune suppression was considered important in the treatment of IPF [4]. In this study, there was no significant difference in the proportion of prednisone prescribed in 34.8% in the non-antifibrotic drug group and 36.0% in the pirfenidone-treated group (p = 0.858). Azathioprine was prescribed for only three patients in the non-antifibrotic drug group, not because IPF, but because of the treatment of inflammatory myopathy developed later. But these cases could not be clearly classified as CTD-ILD or IPF with combined CTD even after case-review. Only N-acetylcysteine had a significantly higher prescription rate in the no-antifibrotic drug group (50.9% vs. 29.0%, p = 0.001).

Although several randomized clinical trials and real-world studies have shown that pirfenidone is efficacious, the doses used were much higher than those employed in South Korea. Pirfenidone doses are often reduced in the real-world due to AEs. As it is unclear whether a lower dose is less effective than a higher one, many clinicians hesitate to prescribe lower pirfenidone doses.

**Table 6. Comparison of annual decline in pulmonary function between patients who were not treated with antifibrotic drugs and patients who were treated with relatively low dose of pirfenidone (600mg/d or 1200mg/d).**

| | Annual decline | | p-value |
|---|---|---|---|
| | No-antifibrotic drug (n = 87) | Treated with relatively low dose of pirfenidone (600mg/d or 1200mg/d) (n = 36) | |
| ΔFVC (L) | -0.328 ± 0.301 | -0.155 ± 0.405 | 0.010 |
| ΔFVC (% predicted) | -9.85 ± 11.432 | -1.245 ± 10.854 | 0.000 |
| ΔFEV1 (L) | -0.219 ± 0.244 | -0.028 ± 0.338 | 0.001 |
| ΔFEV1 (% predicted) | -9.786 ± 13.917 | 3.380 ± 30.921 | 0.001 |
| ΔFEV1/FVC | 1.343 ± 5.841 | 4.471 ± 20.794 | 0.198 |
| ΔDLCO (mL/mmHg/min) | -2.177 ± 2.975 | -0.455 ± 2.873 | 0.004 |
| ΔDLCO (% predicted) | -11.695 ± 16.819 | 2.711 ± 45.085 | 0.011 |

Values are expressed as the mean ± standard deviation (SD).

Abbreviations: Standard deviation, SD; Forced vital capacity, FVC; Forced expiratory volume in one second, FEV1; Diffusing capacity of lung for carbon monoxide, DLCO.

**Table 7. Comparison of annual decline in pulmonary function between patients who were treated with full recommended dose of pirfenidone (1800mg/d) and relatively low dose of pirfenidone (600mg/d or 1200mg/d).**

| | Annual decline | | |
| --- | --- | --- | --- |
| | Treated with relatively low dose of pirfenidone (600mg/d or 1200mg/d) (n = 36) | Treated with full recommended dose of pirfenidone (1800mg/d) (n = 19) | p-value |
| ΔFVC (L) | -0.155 ± 0.405 | -0.081 ± 0.287 | 0.434 |
| ΔFVC (% predicted) | -1.245 ± 10.854 | -2.123 ± 7.683 | 0.730 |
| ΔFEV1 (L) | -0.028 ± 0.338 | -0.042 ± 0.268 | 0.868 |
| ΔFEV1 (% predicted) | 3.380 ± 30.921 | -0.963 ± 10.225 | 0.447 |
| ΔFEV1/FVC | 4.471 ± 20.794 | 1.066 ± 4.062 | 0.348 |
| ΔDLCO (mL/mmHg/min) | -0.455 ± 2.873 | -0.793 ± 2.533 | 0.656 |
| ΔDLCO (% predicted) | 2.711 ± 45.085 | -2.921 ± 13.123 | 0.490 |

Values are expressed as the mean ± standard deviation (SD).

Abbreviations: Standard deviation, SD; Forced vital capacity, FVC; Forced expiratory volume in one second, FEV1; Diffusing capacity of lung for carbon monoxide, DLCO.

In a Japanese phase III clinical trial, both the high-dose (1800 mg/day) and low-dose (1200 mg/day) pirfenidone groups exhibited improved FVC compared to a placebo group [9]. In the CAPACITY trial (Study 004), patients were assigned to a pirfenidone 2403 mg/day, pirfenidone 1197 mg/day, or placebo group in a 2:1:2 ratio. The 2403 mg/day dose was derived by normalizing of the 1800 mg/day dose used in Japanese studies accruing to the predicted body weights of a predominantly US-based population [7]. Pirfenidone at 2403 mg/day significantly reduced the mean decrease in the predicted FVC compared to placebo. The outcomes of the pirfenidone 1197 mg/day group were intermediate between those of the pirfenidone 2403 mg/day and placebo groups [7].

Pirfenidone and another antifibrotic, nintedanib, have become the gold standard for IPF treatment [4]. Pirfenidone is safe and tolerable in the long term. Nevertheless, pirfenidone-related AEs often lead to dose reduction and treatment interruption strategy, and a significant proportion of patients discontinue treatment [11]. In the CAPACITY (Study 004 and Study 006) and ASCEND trials, treatment was discontinued because of AEs in 15% and 14.4% of patients in the pooled pirfenidone groups, respectively [6, 7].

**Table 8. Comparison of incidence of adverse events between patients who were treated with full recommended dose of pirfenidone (1800mg/d) and relatively low dose of pirfenidone (600mg/d or 1200mg/d).**

| | Treated with relatively low dose of pirfenidone (600mg/d or 1200mg/d) (n = 74) | Treated with full recommended dose of pirfenidone (1800mg/d) (n = 26) | p-value |
| --- | --- | --- | --- |
| ≥1 AE of any type | 33 (44.6%) | 24 (92.3%) | 0.000 |
| Poor oral intake | 18 (24.3%) | 7 (26.9%) | 0.792 |
| Nausea, vomiting | 9 (12.2%) | 4 (15.4%) | 0.674 |
| Diarrhea | 6 (8.1%) | 0 (0.0%) | 0.134 |
| Dyspepsia | 5 (6.8%) | 4 (15.4%) | 0.186 |
| Skin rash, itching | 7 (9.5%) | 2 (7.7%) | 0.787 |
| Neurological disorder | 3 (4.1%) | 0 (0.0%) | 0.297 |
| General weakness | 4 (5.4%) | 0 (0.0%) | 0.226 |

Abbreviations: Adverse events, AE.

**Table 9. Comparison of cause of pirfenidone discontinuation between patients who were treated with full recommended dose of pirfenidone (1800mg/d) and relatively low dose of pirfenidone (600mg/d or 1200mg/d).**

| | Treated with relatively low dose of pirfenidone (600mg/d or 1200mg/d) (n = 74) | Treated with full recommended dose of pirfenidone (1800mg/d) (n = 26) | p-value |
|---|---|---|---|
| Discontinuation of any cause | 47 (63.5%) | 11 (42.3%) | 0.059 |
| Death | 8 (10.8%) | 5 (19.2%) | 0.272 |
| Follow up loss | 21 (28.4%) | 3 (11.5%) | 0.084 |
| Adverse events | 11 (14.9%) | 2 (7.7%) | 0.350 |
| IPF progression | 12 (16.2%) | 1 (3.8%) | 0.107 |
| Malignancy | 2 (2.7%) | 1 (3.8%) | 0.769 |

The above-mentioned phase III clinical trials served as the basis for expecting efficacy in relatively low-dose pirfenidone (1200 mg/day). However, the efficacy and safety of lower doses (≤ 1200 mg/day) have not been studied. Also, patients with advanced disease (FVC < 50% or DLCO < 30%) are excluded from clinical trials; real-world studies thus provide more informative data on the efficacy and safety of antifibrotic compounds. In fact, for five (5%) of our patients treated with pirfenidone, the FVC and DLCO were predicted to be below 50% and 30%, respectively. We studied the efficacy of relatively low-dose pirfenidone (≤ 1200 mg/day) in terms of OS and pulmonary function; the low dose was not inferior to the high dose.

Several studies have analyzed the effects of pirfenidone on the survival and all-cause mortality of IPF patients. When the ASCEND and CAPACITY populations were pooled, overall all-cause mortality was shown to be reduced by pirfenidone. At week 52, the all-cause mortality of the pirfenidone 2403 mg/day and placebo groups were 3.5% and 6.7%, respectively (p = 0.01) [6]. During post-hoc analysis of the trial data, patients with advanced IPF (FVC < 50% and/or DLCO < 35%) were analyzed; the all-cause mortality rates in the pirfenidone and placebo groups at week 52 were 4.4% and 15.0%, respectively (HR = 0.28, 95% CI: 0.09–0.86, p = 0.018) [16]. In the RECAP trial (based on the ASCEND and CAPACITY trials), the median survival time from the first dose of pirfenidone (2403 mg/day) was 77.2 months [10]. In a real-world Italian study, the 3-year survival of IPF patients taking pirfenidone was 73% [17]. In the Czech EMPIRE registry, the 60-month OS rates of pirfenidone and no-antifibrotic treatment groups were 0.559 (95% CI: 0.474–0.644) and 0.315 (95% CI: 0.234–0.396), respectively (p = 0.002) [18].

Although there was a difference in that both antifibrotic drugs were included, Cameli et al. including 139 patients treated with pirfenidone and 124 patients treated with nintedanib, according to the study, the median survival was 1224 days during an observational period of 885.3 ± 559.5 days, and there was no significant difference between the two drug groups [19].

In our study, the 1-year all-cause mortality rate was 12.1% in the pirfenidone group (any dose) and 22.9% in the no-antifibrotic drug group; these rates are higher than those in clinical trials. Similarly, the 3-year OS rates were 71.3% and 58.9% in the pirfenidone and no-antifibrotic drug groups, respectively; the respective 5-year OS rates were 57.8% and 42.8%. The mean survival of our pirfenidone-treated group was evaluated over a long period (73.26 ± 7.87 months). Also, most of the pirfenidone-treated patients received only 1200 mg/day of the drug, or less, in contrast to previous real-world studies; the low doses enhanced survival. To the best of our knowledge, no study has compared survival between patients on high- and low-dose pirfenidone.

In the non-antifibrotic drug group, the mean OS was 57.03 ± 3.90 months, which is high compared to other real-world studies. In South Korea, medical costs are relatively low due to the national health insurance. Even if there are no symptoms, low-dose chest CT is performed

every year for lung cancer screening for ever smokers over the age of 55 and 30 pack years. In addition, access to CT is high, including cases where chest CT is performed due to other diseases such as pneumonia, tuberculosis, and COPD. Therefore, patients with IPF are often diagnosed early with no symptoms and preserved lung function. On the other side, the reimbursement criteria for pirfenidone are relatively strict, and eventually many patients with early IPF were included in the group who did not treat with pirfenidone.

In a Japanese clinical trial, the mean decrease in FVC over 52 weeks was 97, 15, and 169 mL in high-dose pirfenidone, low-dose pirfenidone, and placebo groups, respectively [20]. In the ASCEND trial, the FVC had decreased by 164 mL and 280 mL in pirfenidone and placebo groups, respectively, after 52 weeks (p < 0.001) [6]. Post-hoc analysis of patients with advanced IPF (FVC < 50% and/or DLCO < 35%) in the CAPACITY and ASCEND trials revealed that the annual decline in FVC was significantly smaller in the pirfenidone than placebo group (150 vs. 278 mL, p = 0.003) [16].

In a real-world study, Chaudhuri et al. measured the changes in FVC and DLCO each 6 months before and after pirfenidone commencement. The FVC decline changed from -1.043 ± 1.605 to -0.197 ± 0.231, and DLCO decline changed from -1.427 ± 1.568 to 0.1 ± 0.367 [21].

Song et al. showed that low-dose pirfenidone was effective in the real-world; the adjusted mean FVC decreased by 200.7, 88.4, and 94.7 mL in control, low-dose (< 1200 mg/day), and high-dose groups, respectively, in 1 year (p = 0.021) [22]. As in our study, there was no significant difference in the extent of the decrease in FVC between the low- and high-dose groups. However, unlike our study, survival was not analyzed by pirfenidone dose in the previous study.

We found that the annual decreases in FVC and DLCO were significantly smaller in the real-world when the recommended dose of pirfenidone was prescribed, and when the dose was 1200 mg/day or less. The annual decline of FEV1 was also significantly smaller, suggesting that FEV1 could serve as an indicator of pirfenidone efficacy.

In the FIBRONET study, the shorter the difference between the time of IPF diagnosis and the start of antifibrotic treatment, the higher the likelihood that the baseline lung function was preserved and the higher the possibility of relatively stable lung function after 12 months of observation [23].

In this study, most of the AEs involved the GI tract and skin, as in other real-world studies [14, 17, 22, 24]. Song et al. reported that GI AEs, such as dyspepsia, anorexia, and nausea, were significantly more common in low- than high-dose groups [22]. We found that the AE incidence did not vary by dose, but the proportion of patients who experienced at least one AE, of any type, was significantly higher in the group treated with the full recommended dose of pirfenidone (1800 mg/day). That group exhibited a significantly higher BMI compared to the low-dose groups (22.04 ± 3.04, 23.49 ± 2.95 and 24.99 ± 3.81 kg/m$^2$ in the 600, 1200 and 1800 mg/day groups, respectively, p = 0.007); similar trends were reported by other studies [7, 22]. Fang et al. showed that patients with a BMI < 25 kg/m$^2$ were at higher risk of disease progression, acute exacerbation, and death than overweight patients (BMI ≥ 25 kg/m$^2$) [25]. A high BMI was associated with better nutritional status, which enhances the response to pharmacologic treatment and slows the disease course. In our study, although patients given low-dose pirfenidone had a low BMI, neither the OS nor pulmonary function decline were poorer. However, the fact that patients with higher BMI values were more tolerant of high-dose pirfenidone is in line with Fang et al. Thus, in some patients, depending on the BMI, dose reduction may be possible without any reduction in efficacy.

Uehara et al. classified the patients into two groups based on the median value of body surface area (BSA) adjusted dose of pirfenidone (876 mg/m$^2$) [26]. The patient group taking the

higher dose of pirfenidone ($\geq$876 mg/m$^2$) showed a lower decline in lung function ($\Delta$%FVC) compared to the patient group taking the lower adjusted dose (<876 mg/m$^2$). However, a significantly higher BSA-adjusted dose was found in patients with AE, and most patients who discontinued pirfenidone had received a higher dose of pirfenidone. In particular, pirfenidone at medium doses (876–1085 mg/m$^2$) showed a significantly lower annual decline in %FVC than patients taking lower doses, as well as significantly reduced AE, resulting in long term effective treatment. In the present study, there was a significant difference in BSA by pirfenidone dose in patients treated with pirfenidone (1.60 $\pm$ 0.21, 1.67 $\pm$ 0.25, and 1.74 $\pm$ 0.18 m$^2$ in the 600, 1200, and 1800 mg/day groups, respectively, p = 0.037, Table 2). When applying the above criteria to our patients, all patients with a higher BSA-adjusted dose of pirfenidone ($\geq$876 mg/m$^2$) belong to the group receiving 1800 mg/day. But our study is not designed for the issue of BSA and dose and we cannot sure the relationship between BSA-adjusted dose and effectiveness. In South Korea, the dose of pirfenidone per pill is 200 mg, and in actual clinical practice, it isn't easy to adjust the dose between 1200 mg/day and 1800 mg/day. Therefore, we tried to show that pirfenidone can also be used efficaciously and safely at a relatively lower dose of 1200 mg/day or less.

Our study had several limitations. First, it used a retrospective design and was conducted at a single institution. Therefore, selection bias and an influence of unknown confounding factors cannot be ruled out. Second, as this was an observational study, it was difficult to obtain progression-free survival data because only a few pulmonary function tests were repeated, at irregular intervals. Finally, we did not explore acute exacerbations, which could contribute to mortality and may be important during the clinical course of IPF patients.

## Conclusions

Low-dose pirfenidone provided beneficial effects on survival and pulmonary function decline in real-world practice. We suggests that continuation of the medication, even at low doses, can be beneficial for IPF patients. Physicians should consider dose reduction rather than discontinuation if the patient's condition permits this.

## Supporting information

**S1 Data.**
(XLSX)

## Author Contributions

**Conceptualization:** Eung Gu Lee, Yujin Hong, Jiwon Ryoo, Jung Won Heo, Bo Mi Gil, Hye Seon Kang, Soon Seog Kwon, Yong Hyun Kim.

**Data curation:** Eung Gu Lee, Tae-Hee Lee, Yujin Hong, Jiwon Ryoo, Jung Won Heo, Bo Mi Gil, Hye Seon Kang, Soon Seog Kwon, Yong Hyun Kim.

**Formal analysis:** Eung Gu Lee, Tae-Hee Lee, Yujin Hong, Jiwon Ryoo, Jung Won Heo, Bo Mi Gil, Hye Seon Kang, Soon Seog Kwon, Yong Hyun Kim.

**Investigation:** Eung Gu Lee, Tae-Hee Lee.

**Methodology:** Eung Gu Lee, Tae-Hee Lee, Yong Hyun Kim.

**Supervision:** Eung Gu Lee, Yong Hyun Kim.

**Validation:** Eung Gu Lee, Tae-Hee Lee.

**Visualization:** Eung Gu Lee.

**Writing – original draft:** Eung Gu Lee, Yujin Hong, Jiwon Ryoo, Jung Won Heo, Bo Mi Gil, Hye Seon Kang, Soon Seog Kwon, Yong Hyun Kim.

**Writing – review & editing:** Eung Gu Lee, Yujin Hong, Jiwon Ryoo, Jung Won Heo, Bo Mi Gil, Hye Seon Kang, Soon Seog Kwon, Yong Hyun Kim.

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
