## [Decision Letter · Decision Letter 0]

14 Jul 2021

PONE-D-21-12339

Effects of low-dose pirfenidone on survival and lung function decline in patients with idiopathic pulmonary fibrosis (IPF): results from a real-world study

PLOS ONE

Dear Dr. Kim,

Thank you for submitting your manuscript to PLOS ONE. After careful consideration, we feel that it has merit but does not fully meet PLOS ONE’s publication criteria as it currently stands. Therefore, we invite you to submit a revised version of the manuscript that addresses the points raised during the review process.

The reviewers have raised a number of issues around confounding factors with the cohort, medications received, and the like. It is critically important to address these, and to make sure that any changes (and uncertainties around this) are also reflected in the abstract.

We look forward to receiving your revised manuscript.

Kind regards,

James West, PhD

Academic Editor

PLOS ONE

Journal Requirements:

2. We note that you stated that data were collected prospectively. Could you please provide the following additional information in your manuscript Methods:

1) Please state whether data were collected routinely

2) Please state whether treatment with pirfenidone was standard-of-care at your hospital

3) Please state whether dose escalation was managed at the discretion of a treating physician, or for the purposes of research.

3. Please include your tables as part of your main manuscript and remove the individual files. Please note that supplementary tables should be uploaded as separate "supporting information" files.

Reviewers' comments:

Reviewer's Responses to Questions

**Comments to the Author**

1. Is the manuscript technically sound, and do the data support the conclusions?

Reviewer #1: Partly

Reviewer #2: Yes

Reviewer #3: No

2. Has the statistical analysis been performed appropriately and rigorously? 

Reviewer #1: Yes

Reviewer #2: Yes

Reviewer #3: No

3. Have the authors made all data underlying the findings in their manuscript fully available?

Reviewer #1: Yes

Reviewer #2: No

Reviewer #3: Yes

4. Is the manuscript presented in an intelligible fashion and written in standard English?

Reviewer #1: Yes

Reviewer #2: Yes

Reviewer #3: No

5. Review Comments to the Author

Reviewer #1: This interesting paper proposed by Kim et Coauthors is focused on a very interesting area of the complex management of patients with IPF. The study is well designed and results are intriguing, but there are many aspects to be corrected/ameliorated to make the paper suitable for publication in PLOS ONE

1) in the Abstract, you stated that low-dose pirfenidone is the most common prescription in Korea. You should explain this aspect and its motivation in the Introduction

2) real-life evidence of pirfenidone effectiveness is far more extensive (e.g. https://pubmed.ncbi.nlm.nih.gov/33102528/

https://pubmed.ncbi.nlm.nih.gov/33627105/)

3) you should state why the majority of your pateints were not treated with antifibrotic drugs (pre-antifibrotic era? deteriorated clinical status? contraindications?) and if these patients were treated with steroids and/or immunosuppressants and/or NAC. These therapies have been found to be harmful for IPF patients and therefore may influence statistical analysis

4) a median of survival of 57 months in patients untreated is higher than reported by many other real-life studies currently published. You should comment this result in the Discussion

Reviewer #2: Major points:

1. Some medications such as corticosteroid, immunosuppressants affect the progression of IPF or adverse effects such as infection, nausea, anorexia. I suggest that you mention those in Methods.

2. A previous study (Respirology (Carlton,Vic.) 23, 318–324 (2018).) suggested that patients receiving a higher BSA-adjusted dose of pirfenidone showed significantly lower annual decline FVC which was associated with prognosis than those taking lower doses.

In your study, the group in high dose pirfenidone exhibited a significantly higher BMI compared to the low-dose groups.

Those suggest that a group in low dose of pirfenidone with relatively lower BMI had the dose appropriate to the effect. Could you explain that the effect of pirfenidone is based on the dose independently ?

For example, you should mention BSA in study population is homogeneously distributed.

Minor points:

In adverse events section, you didn't mention Table 9 in the document.

In table 9, the frequency of follow up loss in the group in low dose of pirfenidone is significantly higher compared to those in the full recommended dose.

You should better mention the details of follow up loss in Methods or Table, such as transfer to another hospital, lung transplantation, lack of data, and so on.

Reviewer #3: Lee EG, et al. submitted the article studying the benefit of low-dose pirfenidone in patients with IPF. The authors described 295 patients with IPF included from 2008 to 2018. The survival was analyzed with somehow three statistical method; the mean survival time was 57.03 ± 3.90 months in the no-antifibrotic drug group and 73.26 ± 7.87 months in the pirfenidone-treated group (p = 0.027). In the unadjusted analysis, the survival of the patients given pirfenidone was significantly better (hazard ratio [HR] = 0.69, 95% confidence interval [CI]: 0.48–0.99, p = 0.04). After adjusting for age, gender, body mass index, and the GAP score [based on gender (G), age (A), and two physiological lung parameters (P)], survival remained better in the patients given pirfenidone (HR = 0.56, 95% CI: 0.37–0.85, p = 0.006). They also showed the decreases in FVC (%), FEV1% and DLco(%) were significantly smaller (p= 0.000, p = 0.001, and p = 0.007, respectively) in patients given pirfenidone. Thus, they concluded, "Low-dose pirfenidone improved survival and slowed pulmonary function decline in the real-world practice" in the current single-center retrospective cohort study.

It is interesting to investigate the efficacy of low-dose pirfenidone in patients with IPF. However, the current study contains severe problems on study design, logical flow, and generalizability. For instance, in my searching, pirfenidone was launched in 2012 in Korea, whereas the study included the patients from 2008. I do not understand why they included those patients in this time gap other than to increase the study participants, whereas this causes unadjustable bias. Moreover, they did not clarify the inclusion/exclusion criteria. In this aspect, it is unclear why no patients treated with nintedanib, the other antifibrotics in this study, even did not exclude them. As they compared undefined two-time points of pulmonary function tests, the other uncontrolled bias was concerned. Also, if they aimed to compare the patients with low-dose pirfenidone with non-treated or full-dose pirfenidone, it was unclear. In such a biased single-center retrospective study with a relatively small number of low-dose pirfenidone users, their conclusion was totally overstatement.

6. PLOS authors have the option to publish the peer review history of their article (what does this mean?). If published, this will include your full peer review and any attached files.

Reviewer #1: No

Reviewer #2: **Yes: **Manabu Ono

Reviewer #3: No

---

## [Author Response · Author response to Decision Letter 0]

30 Nov 2021

Response to reviewers’ comments

PONE-D-21-12339

Subject: Effects of low-dose pirfenidone on survival and lung function decline in patients with idiopathic pulmonary fibrosis (IPF): results from a real-world study

→ Thank you very much for the giving us an opportunity to revise the manuscript. We have responded to comments raised by reviewers. 

Journal Requirements: When submitting your revision, we need you to address these additional requirements.

1. Please ensure that your manuscript meets PLOS ONE’s style requirements, including those for file naming. The PLOS ONE style templates can be found at 

→ We checked the title, author, affiliations formatting guidelines, and manuscript formatting guidelines of PLOS ONE and edited the manuscript to fit PLOS ONE’s style.

2. We note that you stated that data were collected prospectively. Could you please provide the following additional information in your manuscript Methods:

1) Please state whether data were collected routinely

→ We apologize for not describing enough information about data collection in the methods section. We added the following to the manuscript. (Page 8, Line 155)

Patients’ characteristics (age, gender, smoking status, BMI) and clinical characteristics (medical history, diagnosis, pulmonary function, radiologic patterns, biomarkers) were collected. Clinical and laboratory data, including pulmonary function test and image studies were collected regularly and in real-time at the time of workup and follow up by the pre-set protocols specified for ILD. All the data collected were again retrospectively reviewed. Data on medications were collected throughout the study, including immunosuppressive agents.

2) Please state whether treatment with pirfenidone was standard-of care at your hospital

→ We agree that answer to this comment is necessary. Pirfenidone is used as standard-of-care in our hospital. We added the following to the manuscript. (Page 8, Line 171)

 There is a compulsory and universal health insurance system in South Korea. Pirfenidone is an expensive drug and was approved by the health insurance system in October 2015. The reimbursement criteria for pirfenidone are strict and are limited to patients with a definite IPF based on high resolution CT and/or surgical lung biopsy and with FVC ≤ 90% or DLCO ≤ 80%. Therefore, since then, pirfenidone has been established as the standard-of-care for patients who satisfy the above criteria, and is also used in our hospital. But Nintedanib is not approved to reimburse and is rarely prescribed due to its high cost in South Korea.

3) Please state whether dose escalation was managed at the discretion of a treating physician, or for the purposes of research.

→ We agree that answer to this comment is necessary. We added the following to the manuscript. (Page 9, Line 182)

Dose escalation, dose reduction, or discontinuation of pirfenidone was made at the physician’s discretion, considering the patients’ condition and not for research purposes.

3. Please include your tables as part of your main manuscript and remove the individual files. Please note that supplementary tables should be uploaded as separate “supporting information” files.

→ We sincerely apologize for not following the submission guidelines for tables. We placed each table in the manuscript file directly after the paragraph in which it is first cited. 

Reviewers’ comments:

Reviewer’s Response to Questions

Comments to the Author

1. Is the manuscript technically sound, and do the data support the conclusions?

Reviewer #1: Partly

Reviewer #2: Yes

Reviewer #3: No

2. Has the statistical analysis been performed appropriately and rigorously?

Reviewer #1: Yes

Reviewer #2: Yes

Reviewer #3: No

3. Have the authors made all data underlying the findings in their manuscript fully available?

The PLOS Data policy requires authors to make all data underlying the findings described in their manuscript fully available without restriction, with rare exception (please refer to the Data Availability Statement in the manuscript PDF file). The data should be provided as part of the manuscript or its supporting information, or deposited to a public repository. For example, in addition to summary statistics, the data points behind means, medians and variance measures should be available. If there are restrictions on publicly sharing data-e.g. participant privacy or use of data from a third party-those must be specified.

Reviewer #1: Yes

Reviewer #2: No

Reviewer #3: Yes

4. Is the manuscript presented in an intelligible fashion and written in standard English?

Reviewer #1: Yes

Reviewer #2: Yes

Reviewer #3: No

5. Review Comments to the Author

Reviewer #1: This interesting paper proposed by Kim et Coauthors is focused on a very interesting area of the complex management of patients with IPF. The study is well designed and results are intriguing, but there are many aspects to be corrected/ameliorated to make the paper suitable for publication in PLOS ONE

1. In the Abstract, you stated that low-dose pirfenidone is the most common prescription in Korea. You should explain this aspect and its motivation in the Introduction

→ We found that the description of low-dose pirfenidone in the abstract section was misleading. We sincerely apologize for this. The full recommended dose of pirfenidone in Korea is approved to be 1800 mg/day, and the most prescribed dose is also 1800 mg/day. However, we often have to reduce the dose to 600 mg/day or 1200 mg/day due to poor compliance or adverse effects of pirfenidone. This situation is common in real-world. We studied whether it is effective even at such a relatively low dose. 

Therefore, we modified the following sentence of abstract as follows. (Page 4, Line 68 and Page 4, Line 87)

“However, sometimes it is difficult to use the dose of pirfenidone used in clinical trials.”

“Low-dose pirfenidone provided beneficial effects on survival and pulmonary function decline in the real-world practice.”

2. Real-life evidence of pirfenidone effectiveness is far more extensive (e.g. https://pubmed.ncbi.nlm.nih.gov/33102528/

https://pubmed.ncbi.nlm.nih.gov/33627105/)

→ We appreciate the reviewer’s attentive comment and fully agree with the comment that the real-world eveidence of antifibrotic drugs is far more extensive. We additionally quoted the papers mentioned by the reviewer in the manuscript. (Page 20, Line 412 and Page 22, Line 453)

 “Although there was a difference in that both antifibrotic drugs were included, Cameli et al. including 139 patients treated with pirfenidone and 124 patients treated with nintedanib, according to the study, the median survival was 1224 days during an observational period of 885.3 ± 559.5 days, and there was no significant difference between the two drug groups.”

“In the FIBRONET study, the shorter the difference between the time of IPF diagnosis and the start of antifibrotic treatment, the higher the likelihood that the baseline lung function was preserved and the higher the possibility of relatively stable lung function after 12 months of observation.”

3. You should state why the majority of your patients were not treated with antifibrotic drugs (Pre-antifibrotic era? Deteriorated clinical status? Contraindications?) and if these patients were treated with steroids and/or immunosuppressants and/of NAC. These therapies have been found to be harmful for IPF patients and therefore may influence statistical analysis.

→ We appreciate the reviewer’s attentive comment and fully agree with the comment. 

In South Korea, pirfenidone was approved by the Korean food and Drug Administration in 2012. However, due to the high price and the lack of clinical practice of pirfenidone, it was not included in the health insurance system until October 2015. Among enrolled patients, the majority of patients not treated with pirfenidone were diagnosed in the pre-antifibrotic era. 

In addition, the reimbursement criteria for pirfenidone are strict and are limited to patients with a definite usual interstitial pneumonia pattern on high resolution CT or IPF diagnosed by surgical lung biopsy and with FVC ≤ 90% or DLCO ≤ 80%. Among patients diagnosed with IPF, patients diagnosed earlier than 2015 and did not meet the above pulmonary function criteria because their pulmonary function was preserved cannot be treated with pirfenidone.

 The Official ATS/ERS/JRS/ALAT Clinical practice guideline in 2015 recommended not use combination therapy of N-acetylcysteine, azathioprine, and prednisone in patients with IPF. Previously, immune suppression was considered important in the treatment of IPF. In this study, there was no significant difference in the proportion of prednisone prescribed in 34.8% in the non-antifibrotic drug group and 36.0% in the pirfenidone-treated group (p = 0.858). Azathioprine was prescribed for only three patients in the non-antifibrotic drug group, not because IPF, but because of the treatment of inflammatory myopathy developed later. But these cases could not be clearly classified as CTD-ILD or IPF with combined CTD even after case-review. Only N-acetylcysteine had a significantly higher prescription rate in the no-antifibrotic drug group (50.9% vs. 29.0%, p = 0.001).

 We added the above to the manuscript. (Page 18, Line 352)

4. A median of survival of 57 months in patients untreated is higher than reported be many other real-life studies currently published. You should comment this result in the Discussion

→ We appreciate the reviewer’s attentive comment and fully agree with the comment.

In the non-antifibrotic drug group, the mean OS was 57.03 ± 3.90 months, which is high compared to other real-world studies. In South Korea, medical costs are relatively low due to the national health insurance. Even if there are no symptoms, low-dose chest CT is performed every year for lung cancer screening for ever smokers over the age of 55 and 30 pack years. In addition, access to CT is high, including cases where chest CT is performed due to other diseases such as pneumonia, tuberculosis, and COPD. Therefore, patients with IPF are often diagnosed early with no symptoms and preserved lung function. On the other side, the reimbursement criteria for pirfenidone are relatively strict, and eventually many patients with early IPF were included in the group who did not treat with pirfenidone. 

We added the above to the manuscript. (Page 21, Line 425)

Reviewer #2

Major points:

1. Some medications such as corticosteroid, immunosuppressants affects the progression of IPF or adverse effects such as infection, nausea, anorexia. I suggest that you mention those in Methods.

→ We appreciate the reviewer’s attentive comment and fully agree with the comment. The reviewer’s comment suggested that the following be mentioned in the method, but we apologize for adding it to the discussion in context.

In South Korea, pirfenidone was approved by the Korean food and Drug Administration in 2012. However, due to the high price and the lack of clinical practice of pirfenidone, it was not included in the health insurance system until October 2015. Among enrolled patients, the majority of patients not treated with pirfenidone were diagnosed in the pre-antifibrotic era. 

In addition, the reimbursement criteria for pirfenidone are strict and are limited to patients with a definite usual interstitial pneumonia pattern on high resolution CT or IPF diagnosed by surgical lung biopsy and with FVC ≤ 90% or DLCO ≤ 80%. Among patients diagnosed with IPF, patients diagnosed earlier than 2015 and did not meet the above pulmonary function criteria because their pulmonary function was preserved cannot be treated with pirfenidone.

 The Official ATS/ERS/JRS/ALAT Clinical practice guideline in 2015 recommended not use combination therapy of N-acetylcysteine, azathioprine, and prednisone in patients with IPF. Previously, immune suppression was considered important in the treatment of IPF. In this study, there was no significant difference in the proportion of prednisone prescribed in 34.8% in the non-antifibrotic drug group and 36.0% in the pirfenidone-treated group (p = 0.858). Azathioprine was prescribed for only three patients in the non-antifibrotic drug group, not because IPF, but because of the treatment of inflammatory myopathy developed later. But these cases could not be clearly classified as CTD-ILD or IPF with combined CTD even after case-review. Only N-acetylcysteine had a significantly higher prescription rate in the no-antifibrotic drug group (50.9% vs. 29.0%, p = 0.001).

 We added the above to the manuscript. (Page 18, Line 352)

2. A previous study (Respirology (Carlton, Vic.) 23, 318-324 (2018).) suggested that patients receiving a higher BSA-adjusted dose of pirfenidone showed significantly lower annual decline FVC which was associated with prognosis than those taking lower doses. In your study, the group in high dose pirfenidone exhibited a significantly higher BMI compared to the low-dose groups.

Those suggest that a group in low dose of pirfenidone with relatively lower BMI had the dose appropriate to the effect. Could you explain that the effect of pirfenidone is based on the dose independently? For example, you should mention BSA in study population is homogeneously distributed.

→ We appreciate the reviewer’s attentive comment. 

Uehara et al. classified the patients into two groups based on the median value of body surface area (BSA) adjusted dose of pirfenidone (876 mg/m2). The patient group taking the higher dose of pirfenidone (≥876 mg/m2) showed a lower decline in lung function (Δ%FVC) compared to the patient group taking the lower adjusted dose (<876 mg/m2). However, a significantly higher BSA-adjusted dose was found in patients with AE, and most patients who discontinued pirfenidone had received a higher dose of pirfenidone. In particular, pirfenidone at medium doses (876-1085 mg/m2) showed a significantly lower annual decline in %FVC than patients taking lower doses, as well as significantly reduced AE, resulting in long term effective treatment. In the present study, there was a significant difference in BSA by pirfenidone dose in patients treated with pirfenidone (1.60 ± 0.21, 1.67 ± 0.25, and 1.74 ± 0.18 m2 in the 600, 1200, and 1800 mg/day groups, respectively, p = 0.037, Table 2). When applying the above criteria to our patients, all patients with a higher BSA-adjusted dose of pirfenidone (≥876 mg/m2) belong to the group receiving 1800 mg/day. But our study is not designed for the issue of BSA and dose and we cannot sure the relationship between BSA-adjusted dose and effectiveness. In South Korea, the dose of pirfenidone per pill is 200 mg, and in actual clinical practice, it isn’t easy to adjust the dose between 1200 mg/day and 1800 mg/day. Therefore, we tried to show that pirfenidone can also be used efficaciously and safely at a relatively lower dose of 1200 mg/day or less. 

We have added the above to the manuscript. (Page 23, Line 473)

Minor points:

In adverse events section, you didn’t mention Table 9 in the document.

In table 9, the frequency of follow up loss in the group in low dose of pirfenidone is significantly higher compared to those in the full recommended dose. You should better mention the details of follow up loss in Methods or Table, such as transfer to another hospital, lung transplantation, lack of data, and so on.

→ We appreciate the reviewer’s attentive comment. 

The proportion of patients with follow-up loss showed a tendency to be higher in the group of patients who were prescribed relatively low-dose pirfenidone. Still, the detailed reason could not be identified. 

We have added the above to the manuscript. (Page 16, Line 325)

Reviewer #3:

Lee EG, et al. submitted the article studying the benefit of low-dose pirfenidone in patients with IPF. The authors described 295 patients with IPF included from 2008 to 2018. The survival was analyzed with somehow three statistical method; the mean survival time was 57.03 ± 3.90 months in the no-antifibrotic drug group and 73.26 ± 7.87 months in the pirfenidone-treated group (p = 0.027). In the unadjusted analysis, the survival of the patients given pirfenidone was significantly better (hazard ratio [HR] = 0.69, 95% confidence interval [CI]: 0.48-0.99, p = 0.04). After adjusting for age, gender, body mass index, and the GAP score [based on gender (G), age (A), and two physiological lung parameters (P)], survival remained better in the patients given pirfenidone (HR = 0.56, 95% CI: 0.37-0.85, p = 0.006). They also showed the decreases in FVC (%), FEV1 (%) and DLco (%) were significantly smaller (p = 0.000, p = 0.001, and p = 0.007, respectively) in patients given pirfenidone. Thus, they concluded, “Low-dose pirfenidone improved survival and slowed pulmonary function decline in the real-world practice” in the current single-center retrospective cohort study.

It is interesting to investigate the efficacy of low-dose pirfenidone in patients witih IPF. However, the current study contains severe problems on study design, logical flow, and generalizability. For instance, in my searching, pirfenidone was launched in 2012 in Korea, whereas the study included the patients from 2008. I do not understand why they included those patients in this time gap other than to increase the study participants, whereas this causes unadjustable bias. Moreover, they did not clarify the inclusion/exclusion criteria. In this aspect, it is unclear why no patients treated with nintedanib, the other antifibrotics in this study, even did not exclude them. As they compared undefined two-time points of pulmonary function tests, the other uncontrolled bias was concerned. Also, if they aimed to compare the patients with low-dose pirfenidone with non-treated or full-dose pirfnidone, it was unclear. In such a biased single-center retrospective study with a relatively small number of low-dose pirfenidone users, their conclusion was totally overstatement.

→ Yes, it is right that pirfenidone was launched in 2012 in Korea. But pirfenidone was used late 2015 and used more widely after 2016 in real clinical practice. 

There is a compulsory and universal health insurance system in South Korea. Pirfenidone is an expensive drug and was approved by the health insurance system in October 2015. The reimbursement criteria for pirfenidone are strict and are limited to patients with a definite IPF based on high resolution CT and/or surgical lung biopsy and with FVC ≤ 90% or DLCO ≤ 80%. Therefore, since then, pirfenidone has been established as the standard-of-care for patients who satisfy the above reimbursement criteria and is also used in our hospital. But nintedanib is not approved to reimburse and is rarely prescribed due to its high cost in South Korea. 

We collected all the clinical and laboratory data at the time of diagnosis and follow-up periods according to the pre-set protocols specified for ILD. We have performed pulmonary function tests, HRCT and 6 min walk test every 6 months by the protocols. As a matter of course, some patients did not perform the scheduled test within a preset-time window. But the proportion was low. We hope to consider that the present study was retrospective and had a weakness. 

We have questioned the effectiveness of low dose pirfenidone in patients who cannot tolerate the full dose. In real practice in South Korea, a physician should make a decision to treat with an even lower dose or not treat because we cannot use the other antifibrotics such as Nintedanib.

In this setting, we have 3 groups of patients: antifibrotics naive, low dose pirfenidone, conventional dose pirfenidone. This inevitable situation has forced us to compare 3 groups of patients and to include the patients from 2008. Actually, the treatment pattern was not different in a patient with IPF in the case of with non-antifibrotics between pre-antifibrotic era and post- antifibrotic era, late 2015 in Korea.

---

## [Decision Letter · Decision Letter 1]

9 Dec 2021

Effects of low-dose pirfenidone on survival and lung function decline in patients with idiopathic pulmonary fibrosis (IPF): results from a real-world study

PONE-D-21-12339R1

Dear Dr. Kim,

We’re pleased to inform you that your manuscript has been judged scientifically suitable for publication and will be formally accepted for publication once it meets all outstanding technical requirements.

Kind regards,

James West, PhD

Academic Editor

PLOS ONE

Additional Editor Comments (optional):

Reviewers' comments:

Reviewer's Responses to Questions

**Comments to the Author**

1. If the authors have adequately addressed your comments raised in a previous round of review and you feel that this manuscript is now acceptable for publication, you may indicate that here to bypass the “Comments to the Author” section, enter your conflict of interest statement in the “Confidential to Editor” section, and submit your "Accept" recommendation.

Reviewer #1: All comments have been addressed

Reviewer #2: All comments have been addressed

2. Is the manuscript technically sound, and do the data support the conclusions?

Reviewer #1: Yes

Reviewer #2: Yes

3. Has the statistical analysis been performed appropriately and rigorously? 

Reviewer #1: Yes

Reviewer #2: Yes

4. Have the authors made all data underlying the findings in their manuscript fully available?

Reviewer #1: Yes

Reviewer #2: Yes

5. Is the manuscript presented in an intelligible fashion and written in standard English?

Reviewer #1: Yes

Reviewer #2: Yes

6. Review Comments to the Author

Reviewer #1: The Authors have correctly addressed all the issues; the paper can be published on PLOS ONE as it focused on a interesting topic in ILD setting

Reviewer #2: I admire your efforts of the revision to your manuscript to fulfill the requirement in Review Comments to the Author.

I understand the difficulty in evaluating data about the patients in different backgrounds (before and after pirfenidone covered by health insurance system).

7. PLOS authors have the option to publish the peer review history of their article (what does this mean?). If published, this will include your full peer review and any attached files.

Reviewer #1: No

Reviewer #2: No

---

## [Editor Report · Acceptance letter]

16 Dec 2021

PONE-D-21-12339R1 

Effects of low-dose pirfenidone on survival and lung function decline in patients with idiopathic pulmonary fibrosis (IPF): results from a real-world study 

Dear Dr. Kim:

I'm pleased to inform you that your manuscript has been deemed suitable for publication in PLOS ONE. Congratulations! Your manuscript is now with our production department. 

Kind regards, 

on behalf of

Dr. James West 

Academic Editor

PLOS ONE